# Private Continual Counting of Unbounded Streams

**Ben Jacobsen**
Department of Computer Sciences
University of Wisconsin — Madison
bjacobsen3@wisc.edu

**Kassem Fawaz**
Department of Electrical and Computer Engineering
University of Wisconsin — Madison
kfawaz@wisc.edu

## Abstract

We study the problem of differentially private continual counting in the unbounded setting where the input size $n$ is not known in advance. Current state-of-the-art algorithms based on optimal instantiations of the matrix mechanism [33] cannot be directly applied here because their privacy guarantees only hold when key parameters are tuned to $n$. Using the common 'doubling trick' avoids knowledge of $n$ but leads to suboptimal and non-smooth error. We solve this problem by introducing novel matrix factorizations based on logarithmic perturbations of the function $\frac{1}{\sqrt{1-z}}$ studied in prior works, which may be of independent interest. The resulting algorithm has smooth error, and for any $\alpha > 0$ and $t \leq n$ it is able to privately estimate the sum of the first $t$ data points with $O(\log^{2+2\alpha}(t))$ variance. It requires $O(t)$ space and amortized $O(\log t)$ time per round, compared to $O(\log(n)\log(t))$ variance, $O(n)$ space and $O(n \log n)$ pre-processing time for the nearly-optimal bounded-input algorithm of Henzinger et al. [27]. Empirically, we find that our algorithm's performance is also comparable to theirs in absolute terms: our variance is less than $1.5\times$ theirs for $t$ as large as $2^{24}$.

## 1 Introduction

*Differentially private counting under continual observation* [11, 18] refers to the problem of maintaining accurate running totals over streams of sensitive data. It has attracted a great deal of attention in recent years [3, 4, 6, 7, 14, 16, 23, 27, 28, 35] for its wide-ranging applications in optimization [13, 12, 14, 21, 32, 36], as well as online learning and the private analysis of streaming data more broadly [1, 10, 30, 38, 40]. It has also been used in many large-scale deployments of differential privacy, such as Google's private next-word prediction [34] and Smart Select models [26]. To a large extent, this flurry of activity has been prompted by recent algorithmic improvements in matrix factorizations for streaming data [14, 23, 27, 33], which have dramatically improved privacy/utility tradeoffs compared to classical approaches based on binary trees [11, 18, 29].

One challenge when applying matrix-based algorithms to streaming settings is that they largely assume access to the input size $n$ as a parameter. While this is a very natural assumption in contexts like private model training, researchers from the beginning have also motivated their work with appeals to applications like public health [18] where it is much less clear how the assumption could hold. In fact, it is known that difficulties related to unbounded inputs have influenced practical deployments of DP [2]: famously, after Apple announced that it planned to use differential privacy to collect user data, Tang et al. [39] discovered that their implementation achieved this by guaranteeing privacy only over a single day while allowing privacy loss per-user to grow without bound over time.

Simple solutions to this problem all carry fundamental tradeoffs. Lifting a bounded-input algorithm to the unbounded case with some form of doubling trick [11] asymptotically preserves performance, but it also leads to non-smooth growth in error over time, undermining one of the major selling points of matrix methods over binary trees [27]. Using a bounded matrix mechanism alongside the classical

query model [17, 19] and refusing to produce additional outputs once our privacy budget is exhausted preserves performance and smoothness, but is not truly unbounded and is unlikely to be satisfying in practice. Finally, simple algorithms like adding independent noise to each data point are trivially unbounded with smooth noise, but have extremely sub-optimal error. Our main contribution in this paper is a matrix-based algorithm that overcomes this trilemma: to the best of our knowledge, it is the first private counting algorithm that is simultaneously **smooth** and **unbounded** with **almost-optimal asymptotic error**. It is also practical to implement, with essentially the same complexity as the foundational algorithm of Henzinger et al. [27], and we show empirically that it enjoys small error for realistic input sizes and not just asymptotically.

## 2 Background and Related Work

### 2.1 Background

**Continual (Binary) Counting.** In continual counting [18, 11], the input consists of a (potentially unbounded) stream of data $x_1, x_2, \ldots$. At each time step $t$, the algorithm receives $x_t$ and then produces an estimate of the partial sum $S_t = \sum_{i=1}^{t} x_i$. For the sake of simplicity, we will assume that $x_t$ is a single bit, but our results are easy to generalize to $x_t \in \mathbb{R}^d$.

**Differential Privacy under Continual Observation.** Define two (potentially unbounded) streams $x_1, x_2, \ldots$ and $x'_1, x'_2, \ldots$ to be *neighboring* if there is at most one time step $t$ where $x_t \neq x'_t$. Let $X_n$ (resp. $X'_n$) denote the first $n$ data points in each stream. We say that a mechanism $\mathcal{M}$ satisfies $(\varepsilon, \delta_{priv})$-differential privacy (DP) [20] in the continual observation model [18] if for all neighboring streams $X, X'$, all $n \in \mathbb{Z}^+$, and all measurable events $S$, we have $\mathbb{P}(\mathcal{M}(X_n) \in S) \leq e^{\varepsilon} \mathbb{P}(\mathcal{M}(X'_n)) + \delta_{priv}$. Here, $\mathcal{M}(X)$ represents the full sequence of outputs produced by $\mathcal{M}$ on input $X$. This reduces to the standard definition when the streams are finite; the additional quantification over $n$ is necessary to make sense of the definition for unbounded streams. The mechanisms we study all use additive Gaussian noise, and so can readily be shown to satisfy other notions of indistinguishability as well, such as Rényi DP [37], zCDP [9], or Gaussian DP [15]. By virtue of using Gaussian noise, they also automatically satisfy DP in the stronger *adaptive* setting where $x_t$ is permitted to depend on earlier outputs, which is important for learning applications [14].

Two measures of utility are standard in the literature: the root mean squared additive error and the expected maximum additive error, defined respectively as the maximums over $X_n$ of:

$$\mathrm{err}_n^{\ell_2}(\mathcal{M}) := \sqrt{\frac{1}{n} \mathbb{E}_{\mathcal{M}}\Big(\sum_{t \leq n} (\|S_t - \mathcal{M}(X_t)\|_2^2)\Big)} \quad \mathrm{err}_n^{\ell_\infty}(\mathcal{M}) := \mathbb{E}_{\mathcal{M}}\Big(\max_{t \leq n} \|S_t - \mathcal{M}(X_t)\|_\infty\Big)$$

When $\mathcal{M}$ uses additive Gaussian noise, both expectations and matching high-probability bounds can be derived from $\mathbb{V}(\mathcal{M}(X_t))$, the (non-asymptotic) variance of the noise added as a function of time. This is the main metric we will report when comparing the performance of different algorithms. We additionally say that a mechanism has *smooth error* if $\mathbb{V}(\mathcal{M}(X_t))$ is a smooth function of $t$. This is a desirable property because $\mathrm{err}_n^{\ell_\infty}$ depends on the *maximum* variance across all time steps, which can be significantly higher than the average variance when error is non-smooth [3].

### 2.2 Related Works

*Differential privacy under continual observation* was introduced by Dwork et al. [18] and Chan et al. [11], who independently proposed the *binary mechanism*. Conceptually, this algorithm computes partial sums with the help of a binary tree where leaf nodes contain private estimates of individual data points, and internal nodes contain independent estimates of the subsequences spanned by their children. The binary mechanism of Dwork et al. [18] used Laplacian noise to satisfy $(\varepsilon, 0)$-DP with $\mathrm{err}_n^{\ell_\infty} = O(\log^{2.5}(n))$, which was improved to $O(\log^{2.5}(t))$ by Chan et al. [11]. This was later reduced to $O(\log(t)\sqrt{\log n})$ by Jain et al. [30] by using Gaussian noise instead. A straightforward optimization of the algorithm follows from the observation that at a given time step, it is only necessary to store $\log_2(n)$ nodes of the tree in memory. Subsequent work by Honaker [29] additionally showed that it is possible to reduce the error by roughly a factor of 2 by making efficient use of multiple independent estimates for the same partial sum.

Table 1: Asymptotic Comparison of Private Counting Algorithms

| Mechanism | $\mathbb{V}(\mathcal{M}(X_t)) \cdot \frac{\varepsilon^2}{\log(1/\delta_{priv})}$ | Time for $t$ outputs | Space | Smooth | Unbounded |
|---|---|---|---|---|---|
| Binary [11, 30]* | $O(\log(t)\log(n))$ | $O(t)$ | $O(\log t)$ | No | No |
| Hybrid Binary [11]* | $O(\log^2 t + \log t)$ | $O(t)$ | $O(\log t)$ | No | **Yes** |
| Smooth Binary [3] | $O((\log n + \log\log n)^2)$ | $O(t)$ | $O(\log t)$ | **Yes** | No |
| Sqrt Matrix [27, 23] | $O(\log(t)\log(n))$ | $O(n\log n)$ | $O(n)$ | **Yes** | No |
| **Algorithm 1** | $O(\log^{2+2\alpha} t)$ | $O(t\log t)$ | $O(t)$ | **Yes** | **Yes** |
| **Hybrid Matrix** | $O(\log^2 t + \log^{2+2\alpha}\log t)$ | $O(t\log t)$ | $O(t)$ | No | **Yes** |

*(\*) Originally satisfied $\varepsilon$-DP with Laplacian noise; numbers reported are for the Gaussian variant. A mechanism is smooth if its variance is a smooth function of $t$, and it is unbounded if it doesn't require $n$ as input. Algorithms with bolded names are original to this work. Constant factors differ significantly between algorithms — we investigate this more closely in section 5 and Figure 2.*

Both the original binary mechanism and Honaker's variant have very uneven noise scale across time steps. Recently, Andersson and Pagh [3] proposed a smooth variant of the binary mechanism which equalizes the noise distribution across time steps, further improving $\text{err}_n^{\ell_\infty}$ by a constant factor.

The binary mechanism can be seen as a special case of the general matrix method [33]. Given a dataset $x$ and a matrix $A = LR$, the matrix method estimates $Ax$ by $L(Rx + z) = Ax + Lz$. This satisfies DP when $z$ is a Gaussian scaled to $\|R\|_{1\to2}$, the maximum $\ell_2$ norm over the columns of $R$. The final error also scales with $\|L\|_{2\to\infty}$, the maximum $\ell_2$ norm over the rows of $L$, and so by choosing $L$ and $R$ carefully we can achieve error depending on the factorization norm $\gamma_2(A) = \min\{\|L\|_{2\to\infty}\|R\|_{1\to2} \mid LR = A\}$. This is often much more accurate than calibrating $z$ to $A$ directly. In the case of continual counting, $A = M_{count}$, the all-ones lower-triangular matrix.

This matrix-based approach has rapidly become the standard in the field. Edmonds et al. [22] proved a lower bound of $\Omega(\log n)$ on the per-time-step $\ell_\infty$ error of continual counting by lower-bounding $\gamma_2(M_{count})$. Denisov et al. [14] approached the problem of finding optimal decompositions as a convex optimization problem, building on earlier work in the offline setting [41]. They proved that, for lower-triangular matrices like $M_{count}$, it is sufficient to consider only decompositions where $L$ and $R$ are both lower-triangular. This is a particularly nice result because it implies there is no extra cost for using the matrix method in streaming settings. Subsequently, Fichtenberger et al. [23] provided an explicit factorization $L = R = M_{count}^{1/2}$ and studied its $\text{err}_n^{\ell_\infty}$ error, and Henzinger et al. [27] proved that the same decomposition is simultaneously nearly optimal in terms of $\text{err}_n^{\ell_2}$. More recent work has investigated various computational modifications of this factorization, trading a small amount of accuracy for efficiency by approximating the optimal decomposition in more compact ways, e.g. through finite recurrences [16, 5] or by binning together similar values [4].

A distinct line of work has investigated the unbounded case. In their seminal work, Chan et al. [11] proposed a 'hybrid' mechanism which uses a variant of the classic doubling trick to match the asymptotic $\text{err}_n^{\ell_\infty}$ rate of the binary mechanism when $n$ is unknown. Their technique is generic and can be used to 'lift' any fixed-size algorithm into one that works on unbounded streams with the help of a (potentially inefficient) unbounded algorithm — in section 5 we investigate a version of this mechanism instantiated with Algorithm 1. An alternative strategy for handling unbounded inputs is to only permit queries based on sliding windows or decaying sums [8, 40]. The dual approach, proposed by Bolot et al. [8] and recently extended by Andersson et al. [6], instead relaxes the classical privacy definition to allow DP guarantees for past data points to gradually expire over time.

Our work is most directly inspired by the rational approximation methods of Dvijotham et al. [16], and in particular by their use of the equivalence between certain matrix factorizations and generating functions. While they search for simple approximations of functions to derive computationally efficient factorizations, we take the opposite approach by searching for more complicated functions to derive matrix factorizations that are usable in the unbounded setting. Because we are not directly concerned with reducing computational complexity below $O(n)$ space and $O(n\log n)$ time, we will generally take the $M_{count}^{1/2}$ mechanism of Henzinger et al. [27] and Fichtenberger et al. [23] ("Sqrt Matrix" in Table 1 and Figure 2) as the baseline against which our algorithm is compared.

# 3 Overview of Results

## 3.1 Preliminaries and Notation

The lower-triangular Toeplitz matrix corresponding to a sequence $a_0, a_1, \ldots$ is defined as the square matrix whose $i, j$ entry is $a_{i-j}$ when $i \geq j$ and 0 otherwise. For a given sequence, we will denote the corresponding matrix as $\mathrm{LTToep}(a_0, a_1, \ldots)$. All matrices we consider will be infinite, but for practical computations we will work with $[A]_n$, the leading submatrix formed by the first $n$ rows and columns of $A$. The special matrix $M_{count}$ is defined as $\mathrm{LTToep}(1, 1, 1, \ldots)$.

LTToep matrices are closed under multiplication, and $\mathrm{LTToep}(S) \mathrm{LTToep}(S') = \mathrm{LTToep}(S * S')$, where $S * S'$ denotes the convolution of the sequences $S$ and $S'$. This means that multiplication of LTToep matrices is commutative and can be performed in $O(n \log n)$ time using the Fast Fourier Transform (FFT). Given an analytic function $f(z)$ with Taylor coefficients $a_n = [z^n] f(z)$, we will abuse notation and write $\mathrm{LTToep}(f)$ for $\mathrm{LTToep}(a_0, a_1, \ldots)$. With this notation, applying the Cauchy product to their Taylor series gives us that $\mathrm{LTToep}(f) \mathrm{LTToep}(g) = \mathrm{LTToep}(fg)$. We refer the curious reader to the full version of Dvijotham et al. [16] for an excellent and detailed introduction to LTToep matrices in this context, including proofs of the properties just described.

Finally, we note that we reserve the symbol $\delta$ to refer to the exponent of the $\log \log$ term in Equation 1. When we need to refer to the privacy parameter of a DP algorithm, we will use $\delta_{priv}$.

## 3.2 Main Result

Our main result is the following theorem and its associated algorithm (Algorithm 1):

**Theorem 1.** *For all $\alpha > 0$, there exists an infinite lower-triangular Toeplitz matrix factorization $L, R \in \mathbb{R}^{\infty \times \infty}$ with the following properties:*

- *Joint Validity: For all $n \in \mathbb{Z}^+$, $[L]_n [R]_n = [M_{count}]_n$*

- *Bounded Sensitivity: $\lim_{n \to \infty} \|[R]_n\|_{1 \to 2} = C < \infty$ for some computable constant $C$*

- *Near-Optimal Asymptotic Error: $\|[L]_n\|_{2 \to \infty} = O(\log^{1+\alpha}(n))$*

- *Computability: There exists an unbounded streaming algorithm that at each time step $t$ takes as input $z_t$ and outputs $(Lz)_t$ using $O(t)$ memory and amortized $O(\log t)$ time*

**Corollary 1.** *For all $\varepsilon, \delta_{priv}, \alpha > 0$ there exists an unbounded streaming algorithm for continual counting, described in Algorithm 1, which has the same complexity as in Theorem 1 and satisfies $(\varepsilon, \delta_{priv})$-DP in the continual release model. At each time step $t$, the algorithm adds Gaussian noise with scale $O\big(\log^{1+\alpha}(t) C_{\varepsilon, \delta}\big)$, where $C_{\varepsilon, \delta} = O(\varepsilon^{-1} \sqrt{\log(1/\delta_{priv})})$ is independent of the input.*

We prove Theorem 1 in section 4; Corollary 1 then follows from existing results. [33, 14].

## 3.3 Technical Overview

The matrix decompositions we consider are of the form $L = \mathrm{LTToep}(f(z; -\gamma, -\delta))$, $R = \mathrm{LTToep}(f(z; \gamma, \delta))$, where:

$$f(z; \gamma, \delta) := f_1(z) f_2(z; \gamma) f_3(z; \delta) \tag{1}$$

$$f_1(z) := \frac{1}{\sqrt{1-z}}, \quad f_2(z; \gamma) := \left( \frac{1}{z} \ln \left( \frac{1}{1-z} \right) \right)^\gamma, \quad f_3(z; \delta) := \left( \frac{2}{z} \ln \left( \frac{1}{z} \ln \left( \frac{1}{1-z} \right) \right) \right)^\delta$$

As motivation for why one might study such matrices, recall that the first column of $M_{count}$ corresponds to the Taylor coefficients of $(1-z)^{-1}$. This implies that for any pair of functions $g_1, g_2$ analytic on the open unit disc with product $(1-z)^{-1}$, we have $M_{count} = \mathrm{LTToep}(g_1) \mathrm{LTToep}(g_2)$. Conversely, any non-pathological decomposition of $M_{count}$ into LTToep matrices will correspond to such a pair of functions. So, rather than searching for arbitrary sequences of real numbers (which are complicated to represent and reason about), we can instead take the approach of searching for nice analytic functions with the hope of translating their functional properties into statements about the sequences we're ultimately interested in.

---

**Algorithm 1** Logarithmic Matrix Method

---

**Require:** Matrix parameters $\gamma < -1/2$ and $\delta$, privacy parameter $C_{\varepsilon,\delta}$

    Compute $\Delta \leftarrow \|\text{LTToep}(f(z;\gamma,\delta)\|_{1\to2}$            ▷ See 'Bounded Sensitivity' in section 4

    **for** $t = 1, \ldots, n$ **do**

        Receive input $x_t \in [0,1]$

        Set $S_t \leftarrow \sum_{i=1}^{t} x_t$

        **if** $t = 2^m$ for some integer $m$ **then**

            Sample $z_s \sim \mathcal{N}(0, C_{\varepsilon,\delta}^2 \Delta^2)$ for $t \le s \le 2t-1$

            Compute next $t$ coefficients of $L = \text{LTToep}(f(z;-\gamma,-\delta))$ in $O(t\log t)$ time

                                                         ▷ See 'Computability' in section 4

            Compute next $t$ terms of the sequence $Lz$ in $O(t\log t)$ time with an FFT

        Output $S_t + (Lz)_t$

---

To understand why these functions *in particular* are good candidates, consider the special case where $\gamma = \delta = 0$. This gives us $R = \text{LTToep}((1-z)^{-1/2})$, which is exactly $M_{count}^{1/2}$ [16]. The key issue that makes this sequence inapplicable to unbounded inputs is that it is not square-summable. By Parseval's theorem, this is equivalent to the statement that the function $f$ is not square-integrable. To fix this, we would like to 'nudge' $f$ so that it *becomes* square-integrable while still being as 'close' to the original function as possible. A plausible first attempt in this direction would be to choose $R = \text{LTToep}((1-z)^{-1/2+\alpha})$ for some $\alpha > 0$, but this gives $O(t^\alpha)$ error; the key issue is that $(1-z)^{-\alpha}$ diverges very quickly as $z \to 1^-$ even when $\alpha$ is small. To obtain logarithmic overhead, we require functions like $\ln(1/(1-z))$ that diverge more slowly. Multiplying by $f_2$ turns out to be sufficient to prove Theorem 1, but we show in section 5 that incorporating $f_3$ can further improve our variance by a constant factor. Finally, the inclusion of the $1/z$ and $2/z$ terms in Equation 1 eliminates the 0 of $\ln(1/(1-z))$ at $z = 0$, which ensures that $f$ is always analytic with $f(0) = 1$.

To study the asymptotic growth of $\|[L]_n\|_{2\to\infty}$ and $\|[R]_n\|_{1\to2}$, we draw on the classic work of Flajolet and Odlyzko [24], whose Theorem 3B (reproduced fully in Theorem 2) provides the following asymptotic expansion for the coefficient of $z^n$ in the Taylor series of $f$:

$$[z^n]f(z;\gamma,\delta) \sim \frac{1}{\sqrt{n\pi}}(\ln n)^\gamma (2\ln\ln n)^\delta \cdot \left(1 + O(\ln^{-1}(n))\right)$$

In particular, this implies that if $\gamma = -1/2 - \alpha$ for some $\alpha > 0$, then the series $\sum_{n=0}^{\infty}([z^n]f(z;\gamma,\delta))^2$ converges, which is equivalent to $R$ having bounded column norm. This is the crucial step that allows us to operate on unbounded time streams without the use of doubling tricks — by calibrating our noise scale to this limit, which can be computed exactly by integrating $|f(z;\gamma,\delta)|^2$ over the unit circle using Parseval's theorem, we can guarantee privacy for all finite inputs. Using the same asymptotic expansion, we can also show that the row norm of the corresponding $L$ matrix grows like $O(\log^{1+\alpha}(n))$ when $\delta \ge 0$ or $O(\log^{1+\alpha+o(1)}(n))$ when $\delta < 0$.

Finally, we can compute the first $t$ coefficients of $f_L$ and its convolution with $z$ in $O(t\log t)$ time and $O(t)$ space using FFTs. Algorithm 1 combines all of these ideas together alongside a standard doubling trick on $t$, which allows us to achieve $O(n\log n)$ time complexity even when $n$ is unknown.

## 4 Proof of Theorem 1

We will begin by fixing $\alpha > 0$, and choosing $f_R(z) = f(z; -1/2 - \alpha, \delta)$, $f_L(z) = f(z; 1/2 + \alpha, -\delta)$ with $f$ defined as in Equation 1. Our goal is to show that the matrix decomposition $L = \text{LTToep}(f_L)$, $R = \text{LTToep}(f_R)$ satisfies all four properties listed in Theorem 1.

**Joint Validity.** We have that $LR = \text{LTToep}(f_L f_R) = \text{LTToep}(\frac{1}{1-z}) = M_{count}$.

**Bounded Sensitivity.** We begin by showing that the column norm of $R$ is finite, or equivalently that $f_R \in L^2$. Recall that $\|R\|_{1\to2}^2 = \sum_{n=0}^{\infty}([z^n]f_R)^2$. By Theorem 3B of Flajolet and Odlyzko [24], we know that $([z^n]f_R)^2 = O(n^{-1}\log^{-1-2\alpha}(n)\log^{2\delta}(\log n)) = O(n^{-1}\log^{-1-\alpha}(n))$, and the series therefore converges by the integral test.

To actually compute this sensitivity, we make use of Parseval's theorem, a classic result in Fourier analysis which relates square-summable sequences to functions that are square-integrable over the unit circle. Specifically, it gives us:

$$\sum_{m=0}^{\infty} ([z^m]f_R)^2 = \frac{1}{2\pi} \int_0^{2\pi} |f_R(e^{i\theta})|^2 \, d\theta$$

In Appendix A, we simplify the right-hand side into two integrals over smooth, real-valued functions that can be integrated numerically to high precision.

**Near-Optimal Asymptotic Error.** We first consider the case when $\delta \geq 0$. Once, again Theorem 3B of Flajolet and Odlyzko [24] tells us that $([z^n]f_L)^2 = O(n^{-1}(\ln n)^{1+2\alpha}(\ln \ln n)^{-2\delta}) = O(n^{-1}(\ln n)^{1+2\alpha})$. So, by the definition of big-$O$, there are constants $n_0, C$ such that for all $n > n_0$, we have $\sum_{m=n_0}^{n}([z^m]f_L)^2 \leq C \sum_{m=n_0}^{n} \frac{1}{m}(\ln m)^{1+2\alpha} \leq C \int_{n_0-1}^{n} \frac{1}{x}(\ln x)^{1+2\alpha} \, dx = \frac{C}{2+2\alpha}(\ln(n)^{2+2\alpha} - \ln(n_0-1)^{2+2\alpha})$. Therefore, $\sqrt{\sum_{m=0}^{n}([z^m]f_L)^2} = O(\ln(n)^{1+\alpha})$ as desired.

The case where $\delta < 0$ is similar, except we have $-2\delta > 0$ and so it is no longer true that our coefficients are $O(n^{-1}(\ln n)^{1+2\alpha})$. But for any $\alpha' > \alpha$, $(\ln \ln n)^{-2\delta} = o((\ln n)^{\alpha'-\alpha})$, and so by the same basic argument we can derive an asymptotic error bound of $O(\ln(n)^{1+\alpha+o(1)})$ instead.

**Computability** We formalize the problem we are trying to solve as a game: first, an adversary secretly fixes some integer $n > 0$. Then, at each time step $t = 1, \ldots, n$, our algorithm is required to output $(\text{LTToep}(f_L)z)_t$, where $z \sim \mathcal{N}(0, I_n)$. The algorithm's goal is to achieve an optimal asymptotic dependence on the unknown value $n$ with respect to the total memory and computation used. We will model computation in units of $M(t)$, the cost of multiplying two polynomials of degree $t$. Because polynomial multiplication can be implemented through divide-and-conquer algorithms, we assume that for any $k > 0$, $M(kt) \sim kM(t)$ asymptotically as $t \to \infty$.

Initially, we disregard the challenge of not having access to $n$ and consider the intermediate problem of computing $[\text{LTToep}(f_L)]_t$ for some constant $t$ that we choose ourselves. As a preview of our eventual strategy for handling unbounded inputs, we assume that we have access to a pre-computed solution of size $t/2$. From here, we observe that $\text{LTToep} f(z; -\gamma, -\delta) = \text{LTToep}(f_1(z)) \text{LTToep}(f_2(z; \gamma)) \text{LTToep}(f_3(z; \delta))$. So, it suffices to compute each of these matrices in isolation. The final product can then be computed using $2M(t)$ (or just $M(t)$ if $\delta = 0$).

To compute $\text{LTToep}(f_1(z))$, we use the fact that $f_1 = \sum_{m=0}^{\infty}(-x)^m \binom{-1/2}{m}$. This can be used to derive the recurrence relation $[z^0]f_1 = 1$, $[z^m]f_1 = (1 - \frac{1}{2m})[z^{m-1}]f_1$ presented in prior works [27, 23]. This recurrence lets us compute the coefficients of this matrix in $O(t)$ time and space.

To compute the cofficients of $\text{LTToep}(f_2(z; \gamma))$, we begin with the fact that $[z^m]f_2(z; 1) = \frac{1}{m+1}$, which can be derived from the Taylor expansion of $\ln(1 + z)$. This intermediate sequence can also clearly be computed in linear time and space. To account for the power of $-\gamma$, we use the identity $f_2(z; -\gamma) = \exp(-\gamma \ln(f_2(z; 1)))$. To compute $\ln(f_2(z; 1))$, we use the fact that $\ln'(f_2(z; 1)) = \frac{f_2'(z; 1)}{f_2(z'1)}$. The coefficients of $f_2'(z; 1)$ can be directly computed from the coefficients of $f_2(z; 1)$ through term-by-term differentiation, and using the division algorithm of Hanrot and Zimmermann [25] lets us compute the ratio $f_2'(z; 1)/f_2(z)$ using $2.25M(t)$. Term-by-term integration then allows us to recover all but the constant term of $\ln(f_2(z; 1))$ in linear time, but this is sufficient because we know $\ln(f_2(0; 1)) = \ln(1) = 0$.

Having shown that we can efficiently compute the Taylor coefficients of $-\gamma \ln(f_2(z; 1))$, it remains to find the coefficients of $f_2(z; -\gamma) = \exp(-\gamma \ln(f_2(z; 1)))$. We achieve this by using the algorithm also presented in [25] that takes an order $t/2$ approximation of $\exp(h)$ as input and produces an order $t$ approximation using $2.25M(t)$.

Finally, we can rewrite $f_3(z; \delta)$ as $(\frac{2}{z} \ln(f_2(z; 1)))^\delta$. The allows us to reuse our earlier computation of the Taylor coefficients of $\ln(f_2(z; 1))$. Dividing by $z$ corresponds to a simple shift of the coefficients. All that remains is to account for the power of $\delta$, which can be done in $4.5M(t)$ using the same technique described above for $\gamma$. We can improve this to just $2.25M(t)$ if $\delta = -\gamma$ by computing the quotient $f_2(z; 1)/f_3(z; 1)$ before applying the power operation. Doing so also saves us an extra $M(t)$ by removing the need to multiply $\text{LTToep}(f_2(z; \gamma))$ and $\text{LTToep}(f_3(z; \delta))$ at the end.

In total, we require $11M(t)$ to compute $[L]_t$, plus one more $M(t)$ to compute $[L]_t z$. To extend to the case of general $n$, we initialize $t=2$ and double it when we are asked to output the $t+1$st estimate. This gives us a total cost over the entire input of $12 \sum_{k=1}^{\lceil \log_2 n \rceil} M(2^k) \sim 12 \sum_{k=1}^{\lceil \log_2 n \rceil} 2^{-k+1} M(n) \leq 24M(n)$, which can be cut down to $17.5M(n)$ if $\delta = -\gamma$, or $13M(n)$ when $\delta = 0$. When polynomial multiplication is implemented with FFTs, we arrive at $O(n \log n)$ time and $O(n)$ space complexity, asymptotically matching the algorithm of Henzinger et al. [27]. But note that because we have to adjust $t$ on the fly, our algorithm requires amortized $O(\log t)$ time at each step, compared to $O(n \log n)$ pre-processing and $O(1)$ time per step for Henzinger et al. [27].

## 5    Implementation and Extensions

In this section, we investigate various practical considerations in the implementation of Algorithm 1. We also consider relaxing the assumption that $n$ is completely unknown by giving the algorithm access to (possibly unreliable) side information on its value, and propose an approximate variant (Algorithm 2) which sacrifices some theoretical rigor for a greater than $5\times$ improvement in running time for large inputs. Finally, we show that Algorithm 1 can be used as a subcomponent of the hybrid mechanism of Chan et al. [11] to achieve exactly $O(\log^2 t)$ variance with improved constants compared to their original approach. [1]

**Choosing the value of $\alpha$.**    While it is in principle possible to set $\alpha$ arbitrarily small, this would be unwise. To see why, observe that for the purposes of bounding sensitivity, there is a stark qualitative gap between the matrix LTToep$(f(z; -1/2 - 10^{-100}, \delta))$, which has finite column norm, and the matrix LTToep$(f(z; -1/2, \delta))$ whose column norm diverges. But at the same time, the row norms of the matrices LTToep$(f(z; 1/2 + 10^{-100}, -\delta))$ and LTToep$(f(z; 1/2, -\delta))$ both diverge at pretty much the same rate. This asymmetry implies that reducing $\alpha$ past a certain point does little to improve our actual error while inviting numerical instability.

So, how *should* $\alpha$ be chosen? We observe that, as of mid-2025, the world's population is estimated to be about 8.2e9. If every single one of those people contributed 100 data points to our algorithm, then we would have $\ln^{0.01}(n) \approx 1.03$. We conclude that setting $\alpha = 0.01$ is likely a safe and practical choice for most applications.

**Choosing the value of $\delta$.**    There are three natural choices for setting the value of $\delta$, representing different tradeoffs in speed and accuracy. Setting $\delta = 0$ is the fastest option computationally, but leads to a sub-optimal rate of growth in error. Choosing $\delta = -\gamma$ is roughly $1.5\times$ slower, but substantially improves error when $n$ is greater than 1,000 or so. Finally, the special value $\delta = -6\gamma/5$ is notable because it causes the first two Taylor coefficients of both $f_L$ and $f_R$ to match the function $(1-z)^{-1/2}$ exactly. For a fixed $\gamma$, this gives us the $L$ and $R$ matrices that are closest to $M_{count}^{1/2}$, subject to the intuitively reasonable constraint that $L \succeq R$. This option is the most expensive, but empirically leads to improved error for $n$ greater than $2^{20}$, which is shown in Figure 2.

**Exploiting imperfect information about $n$.**    Up to this point we have assumed that $n$ is unknown and impossible to predict, but in practice, history or expert knowledge might suggest a range of plausible values. We model this side information with the double-inequality $n_0 \leq n \leq Cn_0$ for some $n_0, C > 0$, which is given to the algorithm. We allow these bounds to be unreliable in two senses — the upper bound might be loose, or it might be entirely wrong. We show that our algorithm can make efficient use of this information when it is reliable without incurring any additional error when it isn't.

Our basic strategy is to simply pre-compute $Lz$ out to $Cn_0$ terms. Provided that the upper bound is true, we recover the same $O(n \log n)$ pre-processing time and $O(1)$ work per-iteration as the $M_{count}^{1/2}$ algorithm. Moreover, we suffer no performance cost if the upper bound turns out to be loose because our error at each round depends only on $t$. In contrast, if we used the $M_{count}^{1/2}$ algorithm with input $Cn_0$, we would be forced to calibrate its noise scale to the sensitivity of the larger matrix $[M_{count}^{1/2}]_{Cn_0}$. In the unlucky event where it turns out that $n = n_0$, this would produce a uniform $\sim 1 + \log C/(1 + \log n_0)$ multiplicative scaling of variance across the entire input.

---

[1]Our code is available at https://github.com/ben-jacobsen/central-dpolo/

In the other direction, if the upper bound turns out to be false then our algorithm can recover gracefully using the same doubling trick described in section 4. But by that point, the $M_{count}^{1/2}$ algorithm will have completely exhausted the privacy budget of the first data-point, forcing it to either weaken its privacy guarantees post-hoc or restart entirely with a new estimate for the input size! Even if this new estimate is perfectly correct, the initial mistake will still lead to $O(\log(Cn_0))$ additive error.

**Constant-factor speedups through asymptotic expansions.** Flajolet and Odlyzko [24] derive the following asymptotic expansion for the Taylor coefficients of $f$, which we used earlier in a weaker form to derive the asymptotic error rate of our mechanism:

**Theorem 2** (Paraphrased from Flajolet and Odlyzko [24] Theorem 3B). *Let $\gamma$, and $\delta$ be complex numbers not in $\{0, 1, 2, \ldots\}$. Then the Taylor coefficients of $f(z; \gamma, \delta)$ satisfy:*

$$[z^m]f(z; \gamma, \delta) \sim \frac{1}{\sqrt{m\pi}}(\ln m)^\gamma (\ln \ln m)^\delta \Big(1 + \sum_{k \geq 1} \frac{e_k^{(\gamma,\delta)}(\ln \ln m)}{(\ln(m)\ln(\ln(m)))^k}\Big) \tag{2}$$

*where $e_k^{(\gamma,\delta)}(x)$ is a polynomial of degree $k$:*

$$e_k^{(\gamma,\delta)}(x) = \sqrt{\pi}\frac{d^k}{ds^k}\Big(\frac{1}{\Gamma(-s)}\Big)\Big|_{s=-1/2} \cdot E_k(x)$$

*and $E_k(x)$ is the $k^{th}$ Taylor coefficient of the function $g(u) = (1 - xu)^\gamma \big(1 - \frac{1}{x}\ln(1 - xu)\big)^\delta$.*

While this expression is messy, the cost of computing it is independent of $m$: for fixed approximation order $K$, the derivatives of the reciprocal gamma function can be precomputed and the remaining terms can be calculated in $O(K \log K)$ time using the same techniques described in section 4. This idea naturally suggests Algorithm 2, which switches from exact computations to order $K$ approximations once the relative error of the approximation falls below some threshold $\eta$. We plot relative error as a function of $t, \delta$ and $K$ in Figure 1.

We highlight a potential pitfall with this approach, which is that if we approximate $\hat{L} \approx L$ directly, then we are implicitly choosing $\hat{R} = M_{count}\hat{L}^{-1}$ and can no longer calibrate our noise to $\|R\|_{1\to2}$ as normal. Ideally we would like to be able to prove a bound like $\|M_{count}\hat{L}^{-1}\|_{1\to2} \leq (1 + O(\eta))\|R\|_{1\to2}$, but this appears to be non-trivial. We therefore take the opposite approach, which is to directly approximate $\hat{R} \approx R$ and calibrate our noise to $(1 + \eta)\|R\|_{1\to2}$ with $\hat{L} = \hat{R}^{-1}M_{count}$. This version of the algorithm is provably valid and private, and for large enough inputs it reduces the computational cost at each power of 2 from $12M(t)$ to $2.25M(t)$ (the cost of one series division), closing over $80\%$ of the computational gap between our algorithm and a standard doubling trick. The tradeoff is a multiplicative $(1+\eta)$ increase in error prior to the switch, and the loss of tight, provable bounds on error after the switch. In Figure 1, we compare the performance of the two algorithms, and

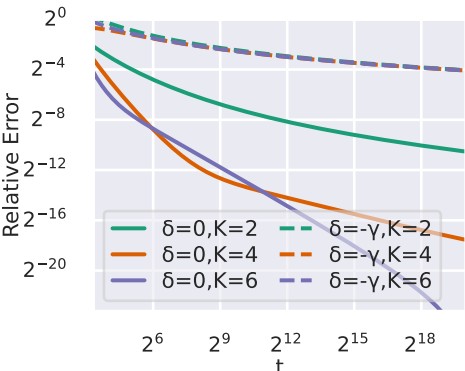

Figure 1: Plots the relative error $|\hat{r}_t - r_t|/r_t$ from Algorithm 2 as a function of $t$ when $\gamma = -0.51$. The asymptotic expansion converges much more quickly when $\delta = 0$.

find that after switching around $t = 2^{11}$, the performance of Algorithm 2 remains indistinguishable from Algorithm 1 out to $t = 2^{24}$.

**Asymptotic improvements through hybrid mechanisms.** In their seminal paper, Chan et al. [11] propose a generic 'Hybrid Mechanism' for continual counting of unbounded streams. The schema requires one (possibly inefficient) unbounded counting algorithm, which operates on the condensed sequence $y_k = \sum_{t=2^k}^{2^{k+1}-1} x_t$, and a bounded algorithm, which is restarted whenever $t$ is a power of 2. The privacy budget is divided between these two algorithms, and the partial sum at time $t = 2^k + r$ is given by adding together estimates for $\sum_{i=0}^{k-1} y_i$ and $\sum_{t=2^k}^{2^k+r} x_t$. The resulting hybrid mechanism is unbounded and typically preserves the exact asymptotic error rate of the chosen bounded algorithm.

---

**Algorithm 2** Approximate Logarithmic Matrix Method

---

**Require:** Matrix parameters $\gamma, \delta \notin \{0, 1, \ldots\}$, order $K$, error tolerance $\eta$, privacy parameter $C_{\varepsilon,\delta}$

    **for** $t = 1, \ldots, n$ **do**
        **if** $t = 2^m$ for some integer $m$ **then**
            Sample $z_s \sim \mathcal{N}(0, (1+\eta)^2 C_{\varepsilon,\delta}^2)$ for $t \le s \le 2t-1$
            $r_t \leftarrow [z^{t-1}]f_R$ using section 4 ('Computability')
            $\hat{r}_t \leftarrow$ Degree $K$ approximation of $[z^{t-1}]f_R$ from Equation 2
            **if** $\frac{|r_t - \hat{r}_t|}{r_t} > \eta$ **then**
                Use section 4 to compute next $t$ entries of LTToep$(f_R)$
            **else**
                Use Equation 2 to *approximate* the next $t$ entries of LTToep$(f_R)$
            Save computed sequence as $[\hat{R}]_{2t}$
            Compute $[\hat{R}]_{2t}^{-1}[M_{count}]_{2t}z$
        Output $(M_{count}\hat{R}^{-1}z)_t$

---

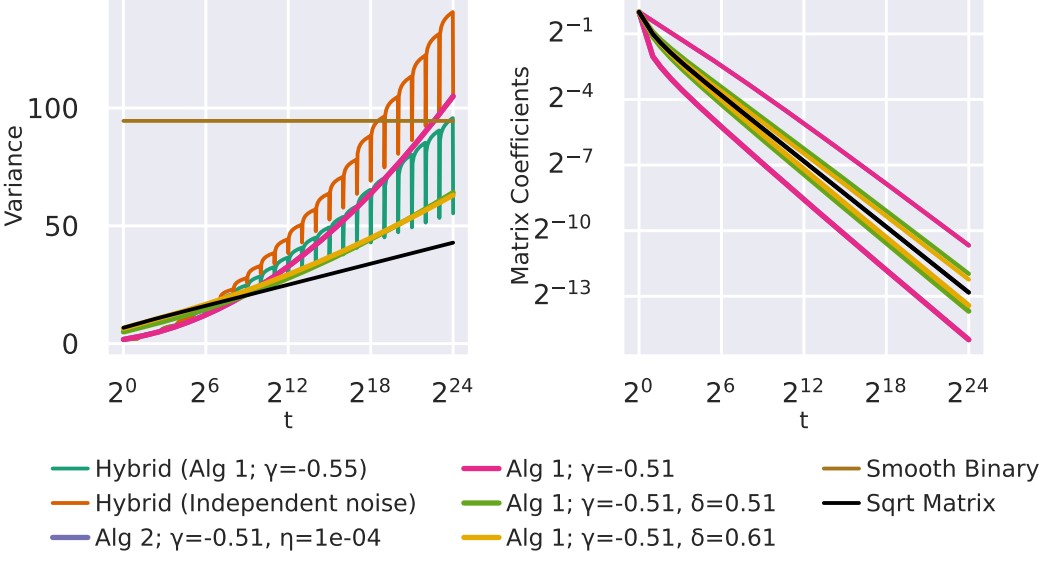

Figure 2: **Left**: Comparison of the exact variance of different algorithms and parameter choices. Contrast with asymptotics in Table 1. The Hybrid mechanism using Algorithm 1 for the unbounded component outperforms the variant using independent noise as in Chan et al. [11], but both variants exhibit very erratic performance when $t$ is close to a power of 2. For Algorithm 1, the parameters $\delta = -\gamma$ and $\delta = -6\gamma/5$ give similar performance, but the latter is slightly better once $t > 2^{20}$, and both outperform $\delta = 0$ when $t > 2^{10}$. Note that the performance of Algorithm 2 with $\delta = 0$ is visually indistinguishable from that of Algorithm 1. **Right**: Coefficients of the matrices LTToep$(f_L)$ (upper) and LTToep$(f_R)$ (lower) for various choices of $\delta$. The choice $\delta = -6\gamma/5$ produces lines that are as close as possible to that of $M_{count}^{1/2}$ without crossing it, which we hypothesize explains its good performance.

The original hybrid mechanism used a simple unbounded algorithm that adds independent noise to each input (corresponding to the matrix decomposition $L = M_{count}, R = I$). Our work enables a more powerful instantiation using Algorithm 1 as the unbounded algorithm instead ('Hybrid Matrix' in Table 1), which we compare against the original. For both implementations, we improve on the original presentation by using $M_{count}^{1/2}$ for the bounded algorithm and allocating 75% of the privacy budget to the bounded learner, which reduces the constant of the leading $O(\log^2 t)$ term. In the spirit of Honaker [29], we also reuse the outputs of the bounded mechanism on earlier subsequences to optimize the variance of the final estimate. Our results are visualized in Figure 2.

# 6 Conclusion

We have shown for the first time that the classic matrix factorization method [33] can be efficiently extended to online settings with unknown input size *without* the use of doubling tricks. The resulting algorithm is the first that we are aware of that is simultaneously **unbounded** and **smooth** with **almost-optimal asymptotic error**. Empirically, we have also shown that it enjoys excellent constants when its parameters are set correctly, with variance that is uniformly less than $1.5\times$ that of Henzinger et al. [27] for inputs as large as $n = 2^{24}$.

Many interesting questions remain unanswered, however. For instance: the only algorithms we are aware of that exactly achieve asymptotically optimal $O(\log^2 t)$ variance in the unbounded setting rely on some form of doubling trick, and despite a great deal of effort, we were not able to find a LTToep decomposition with this property. Our suspicion is that no such decomposition exists, and that this is related to the classic result that the sequence $a_n = (n \cdot \log n \cdot \log \log n \cdot \ldots \cdot \log^{\circ k} n)^{-1}$ diverges for any $k$. We believe that a formal proof of this conjecture would be very interesting as it would constitute a clean separation between the power of LTToep and general lower-triangular matrices in online learning. We therefore conclude by posing the following open question:

**Open Question 1.** *Do there exist* LTToep *matrices* $L, R$ *such that* $[L]_n[R]_n = [M_{count}]_n$, $\lim_{n\to\infty} \|[R]_n\|_{1\to 2} < \infty$, *and* $\|[L]_n\|_{2\to\infty} = \Theta(\log^2 n)$?

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

# A   Technical Details for Computing Sensitivity

To actually compute the sensitivity, we make use of Parseval's theorem, which gives us:

$$\sum_{n=0}^{\infty} ([z^n]f_R)^2 = \frac{1}{2\pi} \int_0^{2\pi} |f_R(e^{i\theta})|^2 \, d\theta$$

$$= \frac{2^{2\delta}}{2\pi} \int_0^{2\pi} (2\sin\left(\frac{\theta}{2}\right))^{-1} \left( \ln^2(2\sin\left(\frac{\theta}{2}\right)) + \frac{(\pi-\theta)^2}{4} \right)^{-1/2-\alpha} \left[ \frac{1}{4} \ln^2 \left( \ln^2(2\sin\left(\frac{\theta}{2}\right)) + \frac{(\pi-\theta)^2)}{4} \right) \right.$$

$$\left. + \left( \text{atan2}\left(\frac{\pi-\theta}{2}, -\ln\left(2\sin\left(\frac{\theta}{2}\right)\right)\right) + \text{atan2}(-\sin\theta, \cos\theta) \right)^2 \right]^{\delta} d\theta$$

Where $\text{atan2}(y, x) = \arctan(y/x)$, possibly shifted by $\pm\pi$ to fall within $(-\pi, \pi]$. With the change of variables $\omega = (\pi - \theta)/2$, this simplifies to:

$$\frac{2^{2\delta}}{\pi} \int_{-\pi/2}^{\pi/2} (2\cos\omega)^{-1} \left( \ln^2(2\cos\omega) + \omega^2 \right)^{-1/2-\alpha} \left[ \frac{1}{4} \ln^2 \left( \ln^2(2\cos\omega) + \omega^2 \right) \right.$$

$$\left. + \left( \text{atan2}(\omega, -\ln(2\cos w)) + \text{atan2}(-\sin(2\omega), -\cos(2\omega)) \right)^2 \right]^{\delta} d\omega$$

From here, we define the functions:

$$I_1(\omega) = 2\cos\omega \quad I_2(\omega) = \ln^2(I_1(\omega)) + \omega^2 \quad I_3(\omega) = \frac{1}{4}\ln^2(I_2(\omega)) \quad I_4(\omega) = \arctan\left(-\omega/\ln(I_1(\omega))\right) + 2\omega$$

By using symmetry and splitting the interval based on the offset of $\text{atan2}$, we finally arrive at:

$$\|R\|_{1\to 2}^2 = \frac{2^{1+2\delta}}{\pi} \left[ \int_0^{\pi/3} I_1(\omega)^{-1} I_2(\omega)^{-1/2-\alpha} (I_3(\omega) + I_4(\omega)^2)^{\delta} \, d\omega \right.$$

$$\left. + \int_{\pi/3}^{\pi/2} I_1(\omega)^{-1} I_2(\omega)^{-1/2-\alpha} (I_3(\omega) + (I_4(\omega) - \pi)^2)^{\delta} \, d\omega \right]$$

Expressed in this way, both integrands are sufficiently smooth to be numerically integrated to high precision. We use mpmath [31] for this purpose in our experiments, which supports arbitrary-precision floating point computations and numerical integration.

