# OpenReview forum: "Private Continual Counting of Unbounded Streams"
_NeurIPS.cc/2025/Conference — NeurIPS 2025 poster_

### Official Review · Reviewer_4GaH · 2025-06-16

**Clarity:** 4
**Significance:** 2
**Originality:** 3
**Rating:** 4
**Confidence:** 3

**Summary:**

This paper addresses the private continual counting problem in the unbounded setting, where the length of the entire data stream is not known in advance. By introducing a novel matrix factorization, the authors present an algorithm that is simultaneously smooth, unbounded, and with almost-optimal error. They also demonstrate through empirical results that their method outperforms previous approaches.

**Questions:**

1. Could the authors provide a concrete illustration of why smoothness is important and what advantages an algorithm with smooth error offers?
2. I am curious why Hybrid Binary and Hybrid Matrix are marked as non-smooth in Table 1. Aren't $\log^2 t+\log t$ and $\log^2t + \log^{2+2\alpha}t$ smooth functions of $t$ ?

**Ethical Concerns:**

["NO or VERY MINOR ethics concerns only"]

**Final Justification:**

This paper addresses the private continual counting problem in the unbounded setting. While the results are nice and the contributions are clear, I am not very sure about the importance of the investigated problem. So I decide to give a borderline accept.

**Limitations:**

yes

**Quality:**

3

**Strengths And Weaknesses:**

Strengths:

1. The authors propose the first algorithm for private continual counting that is smooth, unbounded, and achieves almost-optimal error ($O(\log^{2+2\alpha}(t)))$ at round $t$. The algorithm is based on the matrix mechanism with a novel factorization.
2. They also provide comprehensive experiments, demonstrating that the proposed algorithm outperforms previous methods.

Weakness:

1. The authors do not clearly articulate the benefits of smoothness, which is crucial for understanding the significance of the problem addressed in this paper.
2. The algorithm incurs higher time and space costs compared to binary-based mechanisms.

---

> ### Author Rebuttal · Authors · 2025-07-30
>
> Thank you for your very helpful review! We agree with you and reviewer G47C that we should do a better job of explaining the idea of smoothness and why it's important. We respond to each of your questions and concerns individually below:
>
> **Weakness 1**: We go into more detail about the meaning and importance of smoothness in response to your questions, and we can incorporate these details into the final paper to make the significance of our contribution more clear.
>
> **Weakness 2**: Binary-based methods achieve sub-linear space complexity, but this comes at the cost of much worse constant factors in their error. This is shown in Figure 2, where the Smooth Binary mechanism (brown line) has substantially higher error than both the original square-root matrix method (black line) and our new unbounded method (yellow line).
>
> Because the relative importance of factors like speed, memory usage and accuracy can change a lot depending on the context, we believe that it's valuable to explore the full range of possible tradeoffs. We therefore see our primary contribution as enabling matrix methods to be used in settings where they *would* be practical except for the fact that $n$ is unknown.
>
> **Question 1**: The biggest reason to care about smoothness is that the expected $\ell_\infty$ error generally scales with the maximum variance across all time steps. If that’s a relevant metric in a given context, then allowing your variance to fluctuate a over time is in some sense inefficient - you'll be penalized for the peaks without being rewarded for the valleys. This is also the motivation behind algorithms like Andersson and Pagh's Smooth Binary mechanism, which has comparable $\ell_2$ error to the classic binary mechanism but much lower $\ell_\infty$ error thanks to being smooth.
>
> A secondary concern is that non-smooth error could complicate downstream analysis by introducing artifacts that look like data anomalies. For instance, if we were using DP continual counting to estimate the spread of an infectious disease, then having dramatically different variance from one time step to the next could plausibly lead to panic or overreactions if the output was shared with non-experts.
>
> **Question 2**: We use 'smooth' to describe the actual variance at each time step, and not just its rate of asymptotic growth. If you look at Figure 2, you can see that both hybrid mechanisms have very sharp fluctuations in variance whenever $t$ is close to a power of 2 - this is why they’re categorized as non-smooth. We can clarify this point in the introduction when we first use the term.

---

> > ### Comment · Reviewer_4GaH · 2025-08-06
> >
> > Thank you for your clarification. I will keep my positive score.

---

### Official Review · Reviewer_8hq2 · 2025-06-27

**Clarity:** 3
**Significance:** 3
**Originality:** 3
**Rating:** 5
**Confidence:** 3

**Summary:**

This paper studies the problem of differentially private continual counting for unbounded streams.
In particular, the input is a stream of bits $x_1, x_2, ..., x_n$. At each time step $t$, after receiving $x_t$, the algorithm needs to output a private estimate of the partial sum $\sum_{i=1}^{t} x_i$. Two streams are $x_1, x_2, ..., x_n$ and $x'_1, x'_2, ..., x'_n$ are considered to be neighboring if there is at most one time step $t$ such that $x_t \ne x'_t$.
A stream is said to be unbounded if $n$ is unbounded, which means we do not assume a priori bound on $n$.
A mechanism M is ($\epsilon$, $\delta$)-DP if for all $n = 1, 2, ...$, all neighboring streams $x_1, x_2, ..., x_n$ and $x'_1, x'_2, ..., x'_n$ , and all events $S$, $P(M(x_1, ..., x_n) \in S) \le e^{\epsilon} P(M(x'_1, ..., x'_n) \in S) + \delta$.
The utility of a mechanism is measured from the following aspects in the paper:
- Error: the variance of the noise added at time step $t$
- Time complexity: the amount of time for $t$ outputs
- Space complexity

Another aspect considered in this paper is that if a mechanism is "smooth". A mechanism is smooth if its error is a smooth function of t.

The private continual counting problem is well-studied and has a large body of literature. However, there are various drawbacks of previous algorithms. The matrix mechanisms cannot be directly applied to the unknown n setting, as their privacy guarantee only holds when n is known. The doubling trick leads to suboptimal and non-smooth error.
This paper gets the first algorithm that is simultaneously smooth and unbounded with almost-optimal asymptotic error.
Their technique is built on the rational approximation methods of Dvijotham et al. In particular, they introducing novel matrix factorizations based on logarithmic perturbations of the function $1/\sqrt{1−z}$ .

**Questions:**

- Is the expectation of $M(X_t)$ equal to $S_t$ ? (I guess you need this for $V(M(X_t))$ to be a reasonable error metric)
- How to translate V(M(X_t)) to the more standard error metrics?
- Table 1:
    - Why is Binary Mechanism not smooth while Sqrt Matrix Mechanism smooth, given they have the same $V(M(X_t))$?
    - Is there a typo in the $V(M(X_t))$ of Hybrid Binary Mechanism? Isn't $\log^2(t) + \log(t)$ just $\log^2(t)$?

**Ethical Concerns:**

["NO or VERY MINOR ethics concerns only"]

**Final Justification:**

I think private continual counting in the unbounded setting is an interesting and well-motivated problem. The authors have addressed my questions and I maintain my score.

**Quality:**

3

**Strengths And Weaknesses:**

**Strengths:** This paper obtains nice results for a well-motived and well-studied problem. And it is well-written in general.

**Weaknesses:** The definition of smooth mechanism is confusing to me. It is phrased as an important feature of a mechanism, but it does not seem to be formally defined or explained in detail anywhere. Also see my question below.

---

> ### Author Rebuttal · Authors · 2025-07-30
>
> Thank you for your kind comments! We address all of your questions individually below:
>
> **Question 1**: Yes, all of the mechanisms we consider add zero-mean Gaussian noise, so they're all unbiased estimators.
>
> **Question 2**: To upper-bound the expected $\ell_\infty$ error, there’s a standard trick based on moment-generating functions and the log-sum-exp function; if your mechanism runs for $t$ steps and the maximum variance across time steps is $\sigma^2$, then the expected $\ell_\infty$ error is at most $\sigma \sqrt{2 \log t}$. You can also derive high probability bounds using the Borel-TIS inequality, which essentially says that the maximum of Gaussians has sub-Gaussian tails.
>
> For $\ell_2$ error, linearity of expectation means we can just add up all of the per-time-step variances, take the square root, and normalize. In the case of matrix mechanisms, this gives us error proportional to $\sqrt{\text{tr} LL^T} = \lVert L \rVert_F$. But on that note, your question helped us realize that we had a typo in our original presentation: the $1/n$ factor should be inside the square root, not outside of it. So thank you for prompting us to check over that part of the paper again!
>
> **Question 3**: We use 'smooth' to describe the actual variance at each time step, and not just its rate of asymptotic growth. For the binary mechanism, the noise at time step $t$ depends on the number of '1's in the binary representation of $t$, which makes it oscillate. We actually intended to include the binary mechanism in Figure 2 for comparison, but the oscillations were so extreme that it made the rest of the figure illegible. We can clarify this point in the introduction when we first introduce the term.
>
> **Question 4**: That's true, yes. We present it this way to highlight the differences in the internal structure of the various algorithms - the error of the Hybrid Binary mechanism decomposes into the error of the bounded component (which is $O(\log^2 t)$) and the error of the unbounded component (which is $O(\log t)$).

---

> > ### Comment · Reviewer_8hq2 · 2025-08-06
> >
> > Thanks for the answers! My questions are addressed.

---

### Official Review · Reviewer_nSpn · 2025-06-28

**Clarity:** 3
**Significance:** 3
**Originality:** 3
**Rating:** 4
**Confidence:** 3

**Summary:**

The paper considers matrix factorization mechanism for continually counting elements in a stream $x_1,x_2,\dots$ in a differentially private fashion. At any point of time $t$, our algorithm should return an estimate of $\sum_{i=1}^tx_i$. The paper considers the case where the $x_i$'s are single bits but state that their results generalize easily to the case of arbitrary vectors.

The state-of-the-art methods for countinual counting uses the above matrix factorization mechanism which factors the $n$ by $n$ counting matrix $M$ (lower triangular with $1$s on and below the diagonal) into $M=LR$, and the estimate of the counts are obtained by applying $R$ to the vector $x$ of $n$ updates, adding noise to $Rx$ and then multiplying the final result by $L$. With $L=R=M^{1/2}$ this method is near-optimal for both $\ell_\infty$ error and $\ell_2$ error if I understand the literature correctly.

While powerful, this factorization has the deficit of only working when the length of the stream $n$ is known in advance, which is often an unrealistic assumption. As the length of the stream increases above $n$, the algorithm fails to retain its privacy guarantees. To resolve this, one can apply a doubling trick, but this solution is somewhat ad hoc and unnatural. The paper's main contribution, is to provide a different factorization that works in the limit $n\to\infty$. The error guarantees are quite close to the $\log t \log n$ bound on the variance from Henzinger et al. [27] holding at any time step $t$. Namely at any time step $t$, the variance of their estimator is $\log^{2+2\alpha}t$ for any desired small constant $\alpha$. They also show that the matrix vector products are efficiently computable using the fast Fourier transform.

**Questions:**

Can you detail how you came up with the function $f$? Does a similar construction appear in past work or is this factorization a novel idea? I think the paper was a little unclear about this (or maybe I missed it), but if it's a new construction, that adds to the novelty of your work.

I find that the paper is missing a final statement on the tradeoffs between utility and privacy for continual counting. The table on page 3 does not include any dependencies on $\varepsilon$ and $\delta$. Corollary 1 does mention what the noise scales with in the matrix mechanism, but how does this affect the final bound? I'm assuming such a statement wouldn't be too hard to produce from the stated results, but I think it would benefit the paper. I think presently it's not even mentioned explicitely that you consider approximate DP.

You mention that your results easily generalize to $x_t\in \mathbb{R}^d$. Could you comment on why this is the case. If it generalizes easily, why aren't the theorems stated in this more general setting? Are there any proofs that should be modified and how? Does it affect the proofs that the arriving vectors can have negative coordinates in the general setting?

You compute the variance of your estimator which suffices to get bounds on the expected error. However, often we are interested in also obtaining bounds that work with high probability. Can you discuss whether your techniques achieve this, or whether there is fundamental limitations in getting such bounds?

I think it would be useful to state the theoretical guarantees that are achievable using the doubling trick. I understand that this method has impracticalities, but I'm currently unsure whether your method also comes with asymptotic improvements in the unbounded case. Could you discuss this?


Some more detailed comments:

l136-137. On the contrary, I found this choice unfortunate and very confusing. Is it correctly understood that $\delta$ is used for two completely unrelated notions? It seems so, since later in the paper $\delta$ can be negative.

l175-176: Your work could be more clear about in which regions your functions are analytic. Also, it is not immediately clear to me why the definitions by $z$ makes the functions analytic.

l204 and rest of the page: This proof is quite difficult to read and could use some polishing. Specific comments below:

l206: What is $n_0$? It doesn't seem to be introduced properly and some quantifiers are missing in the statement.

l216-217: This is very informal. I don't know what $M(kt)\sim kM(t)$ means. For $t=1$, with some interpretation, it seems to imply that $M(k)=O(k)$ which I don't think is true.

l227-240: I couldn't  follow how this shows to how we can compute the coefficients as some parts are very loosely written and many details are glossed over. This seems to be one of the core parts of your theoretical contribution, so I think it would be nice to provide more details.

**Ethical Concerns:**

["NO or VERY MINOR ethics concerns only"]

**Final Justification:**

Please see the answer to the authors

**Limitations:**

Yes

**Quality:**

3

**Strengths And Weaknesses:**

(1) I find the paper relatively well motivated in that continual counting is an important problem and it makes sense to strive for a more smooth solution that doesn't resort to 'clumsy' doubling tricks.

(2) It is mathematically quite interesting that there is a matrix factorization mechanism with almost the same error guarantees that works in the limit $n\to \infty$. I also like the open problem at the end of the paper.

\textbf{Weaknesses}:

(1) I find that the technical contribution is perhaps a little underwhelming. The main contribution is the analysis of the new factorization which is technical but relatively straightforward. I am a little in doubt whether the new factorization is new or has appeared in past work, e.g., Flajolet and Odlyzko [24].

(2) While it is nice to have results in the unbounded setting, they come with a worse logarithmic dependence, and it is unclear to me whether their approach would actually lead to better estimators than using some doubling trick.


(3) Parts of the paper are quite hard to read and could use some rewriting for better clarity and mathematical precision.

\textbf{Overall Evaluation}: Taking the above points into consideration I am hesitant to accept the paper at NeurIPS.

---

> ### Author Rebuttal · Authors · 2025-07-30
>
> Thank you for providing so many detailed suggestions and insightful questions! We appreciate being pushed to be more mathematically precise and feel confident that the final paper will be much clearer as a result. We respond individually to each of the weaknesses and questions you raised below:
>
> **Weakness 1**: Although the sequence we consider is based on generating functions that have been studied before, this is also the case for many other works in this area - for instance, Henzinger et al. (2023) use the coefficients of the generating function of $1/\sqrt{1-z}$, which were already known to Newton, and Dvijotham et al. (2024) build on known results about rational approximations of the function $1/\sqrt{1-z}$ on the unit circle that were first published in the 1960s.
>
> Like these prior works, our contribution is not in being the first people to write down this particular sequence, but rather in A) recognizing that the analytic properties of the sequences studied in prior works are equivalent to interesting properties of differentially-private streaming algorithms, like unboundedness, B) showing that the implied algorithm can be implemented in a computationally efficient way, and C) empirically demonstrating that the resulting algorithm is a concrete improvement over the state of the art in relevant metrics.
>
> **Weakness 2**: This is a great point, and it’s true that asymptotically our algorithm is slightly worse than a (well-implemented) doubling trick. But as we show in Figure 2, the slightly-worse logarithmic dependence is counterbalanced by much better constants, and so for the input sizes we evaluate, our approach **does** actually lead to better estimators. In particular, Figure 2 shows that the Hybrid mechanism of Chan et al. (orange line), which is based on a doubling trick, has much higher variance than our proposed algorithm (yellow line).
>
> **Weakness 3**: We address this point in more detail below.
>
>
> **Question 1**: To the best of our knowledge, no prior work in the DP literature has used similar constructions, so in that context we believe it is novel. We also aren’t aware of any earlier work that explicitly considers these functions in the context of LTT matrix factorizations.
>
> As for how we came up with the function $f$ - we'd been working on the general problem for a few months by starting with the square-root factorization, manually perturbing coefficients to make them convergent, and trying to use Fourier analysis to derive features of the implicitly defined function. Reading Dvijotham et al. (2024) put the idea of going in the opposite direction in our heads. Later on, a search for relevant works turned up the Flajolet and Odlyzko paper and we realized that the functions that they had studied in the context of analytic combinatorics could potentially do what we needed.
>
> **Question 2**: Every algorithm we design and compare against satisfies DP by adding Gaussian noise, and so all of them have exactly the same dependence on epsilon and delta. This is why we don't emphasize the dependence on privacy parameters - both our asymptotic results and our comparisons against related work would be unaffected by varying the privacy level. That said, it does make sense that readers would expect to see some sort of dependency on the privacy parameters. To fix this, we'll modify the second column of Table 1 to indicate that it represents the variance *after* being normalized by a data-independent value that depends on epsilon and delta, analogous to Table 1 in Andersson and Pagh (2023).
>
> **Question 3**: Focusing on bits is a convention in the private counting literature, going all the way back to Chan et al. (2011) and Dwork et al. (2010). The general case can be reduced to bit-counting as follows:
>
> First, consider $d$-dimensional bit vectors. We can construct a continual counting algorithm for that setting by applying $d$ versions of our single-bit algorithm in parallel to each coordinate. We can then use composition theorems to control the overall privacy cost, which translates to an extra factor of $d$ in the variance.
>
> Next, consider real-valued vectors in a bounded set with $\ell_\infty$ diameter $D$. By shifting and scaling, we can convert them to vectors in $[0,1]^d$. From there, the fact that they can take intermediate values doesn't impact sensitivity, so our private $d$-dimensional bit counting algorithm will work out of the box. At the end, we rescale our estimates, increasing variance by a factor of $D^2$.
>
> Similar reductions are possible if you instead assume your vectors are in a set with bounded $\ell_2$ diameter, or if they're contained in an ellipsoid. In particular, it doesn't matter if the original coordinates are negative or not - only the distances between them are important. To be more precise, we can revise this section to say that it's easy to generalize our results to vectors in any bounded subset of $\mathbb{R}^d$.
>
> **Question 4**: Yes, high-probability bounds can be derived from the variance of each mechanism. The important point here is that the noise added for privacy follows a zero-mean Gaussian distribution. This means that knowing the variance fully determines the distribution and therefore all relevant error metrics, including both expectations and high-probability bounds.
>
> Concretely, if you want a high probability bound on $\ell_\infty$ error, you can use MGFs to upper bound the expected maximum error across all time steps and then apply the Borel-TIS inequality, which says that the distribution of the maximum of a set of Gaussian random variables has sub-Gaussian tails. Analogous bounds can be derived for $\ell_2$ error using the fact that Borel-TIS-like inequalities hold for any Lipschitz function of a normal random variable, including vector norms (see e.g. equation 5 in "Four Talagrand inequalities under the same umbrella" by Michel Ledoux).
>
> **Question 5**: The most naive doubling trick will give you $O(\log^3 t)$ variance. We don't compare against that algorithm because the Hybrid mechanism of Chan et al. (row 2 of the table 1) is a more sophisticated version of the same idea.
>
> Asymptotically speaking, that algorithm is already optimal (i.e. it achieves $O(\log^2 t)$ variance), but it has pretty poor constants - this is shown in figure 2 (orange line), where it has the worst variance of all of the algorithms we consider. We also show that those constants can be improved substantially by replacing one of its subcomponents with our own algorithm (teal line in figure 2; 'Hybrid Matrix' in table 1). So, even if someone was primarily interested in asymptotic guarantees, they would still be better off using the version of the Hybrid mechanism based on our matrix factorization instead of the classic variant.
>
> **L136-137**: You understand correctly, yes - we're sorry that it ended up being confusing. Our goal here was to make it easier for an interested reader to cross-reference our results with the original Flajolet paper, which uses $\delta$ as the exponent of the $\log\log$ term. We'll revise our definition of DP to use $\varepsilon_{\textit{priv}},\delta_{\textit{priv}}$ instead of $\varepsilon,\delta$ to remove the ambiguity.
>
> **L175-176**: This is a good point and we will clarify - our functions are analytic on the open unit disc. The division by $z$ is there to eliminate the zero of $\ln(1/(1-z))$ at $z=0$, which makes it possible to take further logarithms or raise our function to non-integer powers without running into issues with the domain of definition.
>
> **L206**: Thank you for pointing this out - the constants $n_0$ and $C$ come from the definition of big-$O$, which we will clarify.
>
> **L216-217**: This model of computation comes from the Hanrot and Zimmerman note that we cite and is intended to simplify away linear and logarithmic terms that don’t matter asymptotically. The intuitive justification is that polynomial multiplication can be implemented as a divide-and-conquer algorithm, and so the cost of multiplying polynomials of degree $tk$ is asymptotically equal to the cost of multiplying $k$ pairs of polynomials with degree $t$.
>
> The upshot of using this model of computation vs. big-$O$ is that it makes it easy to track the multiplicative constants - every parameterization of our algorithm requires $O(n \log n)$ time, but this approach lets us additionally estimate that choosing $\delta=0$ should roughly halve the actual computation required, for example.
>
> Regardless, your counterexample definitely highlights that we were too informal in our presentation, and we'll add a clarification that $M(kt) \sim kM(t)$ should be interpreted as holding asymptotically for large $t$.
>
> **L227-240**: This section is definitely very dense. Given the suggestions from other reviewers about the clarity of figures and spots where we could include more detail, we plan to move the technical exposition here to the appendix. This will give us space to make it more explicit by giving a step-by-step formula for constructing each of the sequences we consider instead of describing the process in natural language.

---

> > ### Comment · Reviewer_nSpn · 2025-08-04
> >
> > Thanks for your detailed feedback, your points make sense to me, and I think I was perhaps too critical in assessing the novelty of the techniques, especially since you state that you are not aware of earlier work using these functions for LTT matrix factorization. I will update my score accordingly.
> >
> > By the way, I think it is not a problem to consider factorizations known from past work, and they are interesting to apply in the present setting. However, it was a little unclear to me where these factorizations came from, and I think it would be great to be precise about this, since there *is* a big difference between finding them in past work and coming up with them yourself. But it's possible I've missed it, and you do state this clearly somewhere in the manuscript.

---

### Official Review · Reviewer_G47C · 2025-06-30

**Clarity:** 4
**Significance:** 4
**Originality:** 4
**Rating:** 5
**Confidence:** 3

**Summary:**

This paper studies the problem of privately counting streaming data when the input size, $n$ (the total number of observations), is unknown. The state-of-the-art approach is the matrix mechanism, but it is only effective when $n$ is known in advance. The authors propose an algorithm that operates in the unbounded setting (where n is not known beforehand) and achieves a smooth error over time with an almost-optimal asymptotic error.

**Questions:**

1. Can the authors clarify the privacy implications of selecting $\alpha$?

2. I think the unbounded setting is relevant in practice. I am more skeptic of how important smoothness is? I am not familiar with specific applications but in many cases (e.g. SGD, or counting) one does not care much about results after a burn-in period. Can the authors comment on this?

3. It seems like this algorithm is good when the time is unbounded (potentially large, otherwise one could set up a cap and use previous algorithms) but in this long running times setting memory and computation of the algorithm seem large. Is there hope to improve on this? could this be a limiting factor for certain online applications?


4. For Algorithm 2 , after switching to the approximation, you lose bounds on error. While it performs well empirically out to t=2^24, can you provide any intuition or heuristic bounds on how the error might behave in a worst-case scenario after the switch?

**Ethical Concerns:**

["NO or VERY MINOR ethics concerns only"]

**Final Justification:**

I am maintaining my score, the authors appropriately answered my questions. I would suggest they clarify the importance of smoothness in this context, since multiple reviewers were initially unclear on the definition and importance.

**Limitations:**

The presentation of the experimental results could be significantly improved. The figures are difficult to interpret due to unclear labels and uninformative captions.

It was challenging to distinguish between the different plotted lines, such as the baselines versus Algorithm 1 and Algorithm 2. Furthermore, the specific parameters used for each result were not clearly indicated within the figures themselves. The captions lacked sufficient detail, forcing me to jump back and forth between the main text in Section 5, the figures, and the captions to understand the setup and outcomes.

**Quality:**

3

**Strengths And Weaknesses:**

**Strenghts**

The paper studies a significant problem in the literature: counting unbounded streams of data. The paper is clearly written. It relies on previous work but proposes an elegant solution to avoid the doubling trick.

The core idea is to use infinite matrices while controlling the column norm of the factorization matrix R, so that one does not need access to the total count n. The authors achieve this by cleverly designing the underlying function f, incorporating the parameters α and γ, and relying on previous work that analyzes the norm of corresponding matrices.

**Weaknesses**

- Throughout the introduction and theorem statements, the purpose of $\alpha$ is completely obscure besides its contribution to the error. This is not explained until Section 3, where the authors clarify its role. They add a third function to make the function f square-integrable, which prevents the error from blowing up by $t^\alpha$  while ensuring the function diverges slowly as z→1.

---

> ### Author Rebuttal · Authors · 2025-07-30
>
> Thank you for your review! It was very helpful and we appreciated all of the insightful questions. We address each of them individually below:
>
> **Question 1**: As long as you pick $\alpha>0$, $R$ will have a bounded matrix norm, so from a privacy perspective the exact value doesn't matter that much - we can always compute the sensitivity and add sufficient noise to guarantee whatever level of privacy we want. At a high level, we think that selecting $\alpha$ is best understood as a (fairly low-impact) tradeoff between constant factors and asymptotic growth rates in the variance.
>
>
> **Question 2**: The biggest reason to care about smoothness is that the expected $\ell_\infty$ error generally scales with the maximum variance across all time steps. If that’s a relevant metric in a given context, then allowing your variance to fluctuate erratically over time is in some sense inefficient - you'll be penalized for the peaks without being rewarded for the valleys. This is also the motivation behind algorithms like Andersson and Pagh's Smooth Binary mechanism, which has comparable $\ell_2$ error to the classic binary mechanism but much lower $\ell_\infty$ error thanks to being smooth.
>
> A secondary concern is that non-smooth error could complicate downstream analysis by introducing artifacts that look like data anomalies. For instance, if we were using DP continual counting to estimate the spread of an infectious disease, then having dramatically different variance from one time step to the next could plausibly lead to panic or overreactions if the output was shared with non-experts.
>
> **Question 3**: We agree that this is an important consideration for very large-scale deployments. If we restrict ourselves to LTT matrix factorizations, then there isn’t any hope to improve on the space complexity: recent work by Andersson and Yehudayoff has shown that for LTT matrices like ours that correspond to irrational functions, $\Omega(t)$ space complexity is unavoidable, and generating noise with LTT matrices is fundamentally tied to convolution so we would be very surprised if it was possible to do better than $O(t \log t)$ time.
>
> The Toeplitz part of the LTT restriction isn’t fundamental, however, and we’re aware of another recent work in this area (Andersson and Pagh, 2025, SaTML) which uses binning to approximate Toeplitz matrices in a way that only requires sub-linear space. One potential obstacle we can foresee with directly applying their results to our factorization is that their algorithm assumes efficient random access to the $L$ matrix being approximated, which isn’t obviously possible in our case. But, we speculate that this could maybe be circumvented for sufficiently large inputs by using the asymptotic expansion for the elements of $L$ alongside some robustness analysis to show that their algorithm can still work well if the input is allowed to have small errors.
>
> **Question 4**: In general, we believe our empirical results are the most compelling evidence that the approximation is 'safe' from a utility perspective. But as a very rough heuristic, if we imagine that Equation 2 systematically had relative error $\eta$ across the board, then the inverse matrix would have relative error of roughly $\eta$ as well, which would imply something like a $(1+\eta)^2$ overhead in error after adjusting for sensitivity.
>
> In practice we see that the relative error varies over time, but it does so in a smooth, monotonic way. In our view, it would be pretty surprising if that led to the behavior of the inverse matrix being qualitatively very different from the idealized case of uniform error, especially if that qualitative difference only emerged after a very long period in which it behaved exactly as we would expect.
>
> **Limitations**: Thank you for the helpful feedback! We aren’t totally satisfied with that figure either and have a few ideas for improvements. In particular, for the final version we would like to split the two subfigures apart, which would make it easier to distinguish the lines and also give us more horizontal space to include longer and more informative labels in the legend. We expect that we’ll be able to find enough extra space for this by moving the more technical components of the Computability proof in Section 4 to the appendix, which will also let us expand that section in line with reviewer nSpn’s suggestions. We can also take another pass over the captions to make them more self-contained so readers won’t have to do so much jumping around.

---

> > ### Comment · Reviewer_G47C · 2025-08-05
> >
> > Thanks for the clear rebuttal!

---

### Official Review · Reviewer_mDPS · 2025-07-02

**Clarity:** 3
**Significance:** 3
**Originality:** 3
**Rating:** 5
**Confidence:** 3

**Summary:**

This paper introduces a new differentially private algorithm for the continual release of running counts over streams of unbounded length. Their work addresses a major limitation of prior methods that assume a fixed or bounded stream size. The algorithm achieves near-optimal error (variance O(\log^{2 + 2\alpha} t)) using space O(t) and amortized time O(log(t)).
The authors construct a novel matrix mechanism by factoring the prefix-sum matrix into two infinite lower-triangular Toeplitz matrices, using carefully designed generating functions. These functions are modified with logarithmic terms to ensure the resulting matrix has bounded column norms as otherwise, the sensitivity of the streams would be unbounded.
The authors also integrate their mechanism into the hybrid framework of Chan et al., replacing bounded components with their unbounded version to achieve smoother and more accurate continual counts. They further propose an approximation variant using asymptotic expansions for faster practical performance.

**Questions:**

1) Line 182 states that the algorithm operates on unbounded streams without the use of doubling tricks, but Line 189 states that you use a standard doubling trick — which is it?

2) Table 1 is confusing, what is the Hybrid Matrix? Is this your contribution? Otherwise where is the citation?

**Ethical Concerns:**

["NO or VERY MINOR ethics concerns only"]

**Final Justification:**

The authors have responded to my concerns, and I have raised my score accordingly.

**Limitations:**

yes

**Quality:**

3

**Strengths And Weaknesses:**

Strengths

This paper addresses an important open question in the area of continual counting. The techniques in this paper are strong and builds upon the intuition of previous works

Weaknesses

The paper would benefit from additional technical details and preliminary material to help readers understand the proposed techniques. At minimum, the formal statements of cited theorems (e.g., Theorem 3B) should be included in the appendix for completeness.
Corollary 1 should include an explicit proof, especially since this represents the paper's main result. Simply stating that it follows from prior work makes it difficult for readers unfamiliar with those references to verify the correctness of the claim. Providing a complete derivation would make the paper much more accessible.

---

> ### Author Rebuttal · Authors · 2025-07-30
>
> Thank you for the very helpful review! We agree with many of your suggestions and address all of the weaknesses and questions you raised below:
>
> **Weaknesses**: We definitely want our paper to be accessible to as broad an audience as possible, so this is very valuable feedback! We are happy to include both a formal statement of all cited theorems and a formal proof of Corollary 1 as additional appendices in the final version. The essential idea of the proof is described when we introduce matrix methods in the Related Work section (line 85), but we can make this more explicit.
>
> **Question 1**: We **don't** use a doubling trick in the sense of restarting the entire private learning algorithm, which would lead to a lot of excess error.
>
> We do, however, use a doubling trick when precomputing matrix products since it lets us amortize the computational cost. This is only relevant for speed and not for accuracy. We agree that the use of two different senses of the term ‘doubling trick’ so close together is unfortunate, and we’ll edit the indicated lines to remove the ambiguity.
>
> **Question 2**: Hybrid Matrix is the Hybrid schema of Chan et al. instantiated with our unbounded matrix mechanism, so it's original to our work. To improve clarity, we'll modify the table to add a reference to the end of Section 5 where it's defined, and also edit that section to explicitly flag the fact that the algorithm discussed is the same as the 'Hybrid Matrix' in the table.

---

> > ### Comment · Reviewer_mDPS · 2025-08-05
> >
> > Thanks for the clear rebuttal. I will make sure to update my score accordingly.

---

### Decision · Program_Chairs · 2025-09-17

**Decision:**

Accept (poster)

**Comment:**

This paper studies differentially private algorithms for the continual release of running counts over streams of unbounded length, observing that previous works assume a fixed or bounded stream size. The algorithm achieves near-optimal error by constructing a novel matrix mechanism that factors the prefix-sum matrix into two infinite lower-triangular Toeplitz matrices, using carefully designed generating functions.

Despite the high scores of the reviews, concerns emerged about the motivation of the problem setting of unbounded settings, as the abstract assumes $t\le n$ and whether the technical novelty of the paper is sufficiently interesting given that there exist standard doubling tricks, in particular, given that both the matrix mechanism and binary tree mechanism are standard approaches for continual release for private counting. Nevertheless the reviewers agreed that the quantitative contribution of the paper is valuable.

The reviewer feedback includes a number of actionable items that could strengthen the potential impact of this work; I would encourage the authors to take these points into consideration.